# Higher Preoperative Maximum Standardised Uptake Values (SUV_max_) Are Associated with Higher Biochemical Recurrence Rates after Robot-Assisted Radical Prostatectomy for [^68^Ga]Ga-PSMA-11 and [^18^F]DCFPyL Positron Emission Tomography/Computed Tomography

**DOI:** 10.3390/diagnostics13142343

**Published:** 2023-07-11

**Authors:** Katelijne C. C. de Bie, Hans Veerman, Yves J. L. Bodar, Dennie Meijer, Pim J. van Leeuwen, Henk G. van der Poel, Maarten L. Donswijk, André N. Vis, Daniela E. Oprea-Lager

**Affiliations:** 1Department of Urology, VU University, Amsterdam University Medical Centers, De Boelelaan 1117, 1081 HV Amsterdam, The Netherlands; h.veerman@amsterdamumc.nl (H.V.); y.j.bodar@amsterdamumc.nl (Y.J.L.B.); d.meijer2@amsterdamumc.nl (D.M.); a.vis@amsterdamumc.nl (A.N.V.); 2Department of Radiology & Nuclear Medicine, VU University, Amsterdam University Medical Centers, De Boelelaan 1117, 1081 HV Amsterdam, The Netherlands; d.oprea-lager@amsterdamumc.nl; 3Prostate Cancer Network The Netherlands, 1066 CX Amsterdam, The Netherlands; h.vd.poel@nki.nl; 4Department of Urology, Antoni van Leeuwenhoek Hospital—The Netherlands Cancer Institute, Plesmanlaan 121, 1066 CX Amsterdam, The Netherlands; 5Department of Radiology and Nuclear Medicine, Antoni van Leeuwenhoek Hospital—The Netherlands Cancer Institute, Plesmanlaan 121, 1066 CX Amsterdam, The Netherlands; m.donswijk@nki.nl

**Keywords:** prostate cancer, standardised uptake value, biochemical recurrence, [^68^Ga]Ga-PSMA-11, [^18^F]DCFPyL, PET/CT

## Abstract

This study aimed to investigate the association between the ^68^Ga- or ^18^F-radiolabeled prostate-specific membrane antigen (PSMA) tracer expression, represented by the maximum standardised uptake value (SUV_max_) of the dominant intraprostatic lesion, and biochemical recurrence (BCR) in primary prostate cancer (PCa) patients prior to robot-assisted radical prostatectomy (RARP). This was a retrospective, multi-centre cohort study of 446 patients who underwent [^68^Ga]Ga-PSMA-11 (*n* = 238) or [^18^F]DCFPyL (*n* = 206) Positron Emission Tomography/Computed Tomography (PET/CT) imaging prior to RARP. SUV_max_ was measured in the dominant intraprostatic PCa lesions. [^18^F]DCFPyL patients were scanned 60 ([^18^F]DCFPyL-60; *n* = 106) or 120 ([^18^F]DCFPyL-120; *n* = 120) minutes post-injection of a radiotracer and were analysed separately. To normalise the data, SUV_max_ was log transformed for further analyses. During a median follow-up of 24 months, 141 (30.4%) patients experienced BCR. Log_2_SUV_max_ was a significant predictor for BCR (*p* < 0.001). In the multivariable analysis accounting for these preoperative variables: initial prostate-specific antigen (PSA), radiologic tumour stage (mT), the biopsy International Society of Urological Pathology grade group (bISUP) and the prostate imaging and reporting data system (PI-RADS), Log_2_SUV_max_ was found to be an independent predictor for BCR in [^68^Ga]Ga-PSMA-11 (HR 1.32, *p =* 0.04) and [^18^F]DCFPyL-120 PET/CT scans (HR 1.55, *p =* 0.04), but not in [^18^F]DCFPyL-60 ones (HR 0.92, *p =* 0.72). The PSMA expression of the dominant intraprostatic lesion proved to be an independent predictor for BCR in patients with primary PCa who underwent [^68^Ga]Ga-PSMA-11 or [^18^F]DCFPyL-120 PET/CT scans, but not in those who underwent [^18^F]DCFPyL-60 PET/CT scans.

## 1. Introduction

Prostate cancer (PCa) is the second most commonly diagnosed cancer among men around the world [1]. Due to the growing and increasingly aging population, a further rise in PCa cases can be expected [2]. This stresses the importance of accurate disease staging and the subsequently adequate treatment of PCa. Unfortunately, a substantial proportion of men experience the biochemical recurrence (BCR) of a disease after treatment with a curative intent [3]. In patients with localised PCa who underwent robot-assisted radical prostatectomy (RARP), BCR is seen in approximately 30% of patients within 10 years after surgery [4]. The current models predicting the risk of recurrence after surgery are based on both clinical and pathological variables, as well as on imaging techniques, such as multiparametric magnetic resonance imaging (mpMRI) [3,5].

Recently, Prostate-Specific Membrane Antigen Positron Emission Tomography/Computerized Tomography (PSMA PET/CT) was introduced as a novel imaging technique, improving the prediction of the disease outcome among patients with PCa compared to that of conventional imaging techniques [6]. In current practice, PSMA is usually labelled with ^68^Gallium (e.g., [^68^Ga]Ga-PSMA-11) or ^18^Fluorine (e.g., [^18^F]DCFPyL). PSMA PET/CT labelled with either ^68^Gallium or ^18^Fluorine has widely proven its prognostic value among patients with recurrent PCa and is therefore implemented in the follow-up of PCa [3,7]. However, there are some essential differences between the two tracers. The use of ^18^Fluroine offers an improved PET image resolution compared to that of ^68^Gallium thanks to its shorter positron range and higher positron yield [8]. Also, ^18^Fluorine has a higher binding affinity to PSMA receptors [8], potentially enhancing the detection of small metastases. Moreover, ^18^Fluorine is a cyclotron product with a longer half-life of 110 min compared to that of ^68^Gallium, a generator product with a shorter half-life of 68 min. The cyclotrons on site enable the centralised and large-scale production of ^18^Fluorine-radiolabelled PSMA tracers, potentially making them more cost-effective and suitable for clinical applications.

In line with the European Association of Nuclear Medicine (EANM) standardised reporting guidelines for PSMA-PET (E-PSMA), the semi-quantitative measurement of PSMA tracer uptake, expressed as maximum Standardised Uptake Values (SUV_max_), is one important element when interpreting PET/CT scans [9,10]. Previous studies suggest that the SUV_max_ of the primary prostate tumour may predict unfavourable disease outcomes, such as a higher pathological International Society of Urological Pathology Grade Group (pISUP), pathological tumour stage (pT) and lymph node involvement [11,12,13]. Furthermore, research showed that the SUV_max_ in the primary staging of PCa represents an independent predictor for disease recurrence with the radiotracer [^68^Ga]Ga-PSMA-11 [11,14,15]. However, the predictive value of PSMA expression with [^18^F]DCFPyL for recurrent disease is yet unknown.

Therefore, the aim of this research was to investigate whether tumour PSMA expression, calculated as SUV_max_, on PSMA PET/CT for staging PCa contributes to the preoperative and postoperative predictions of BCR occurrence after curative treatment.

## 2. Materials and Methods

### 2.1. Study Design and Patient Population

This was a retrospective, multi-centre cohort study. Patients with histologically proven PCa who underwent RARP between August 2016 and June 2022 were included. All patients underwent [^68^Ga]Ga-PSMA-11 PET/CT or [^18^F]DCFPyL PSMA PET/CT prior to RARP, with or without extended pelvic lymph node dissection (ePLND). An ePLND was performed based on a ≥7% risk of lymph node involvement, as calculated using the Briganti 2012 nomogram or the Memorial Sloan Kettering Cancer Center (MSKCC) nomogram [16,17]. Patients were excluded if the preoperative PSMA PET/CT showed no focal PSMA uptake (non-PSMA-expressing tumour [miT0]) [10,18], if the serum PSA was not available postoperatively, if the patients underwent prior treatment for PCa, or if distant metastases were found on preoperative PSMA PET/CT.

Patients were included in two reference centres of the Prostate Cancer Network Netherlands, the Antoni van Leeuwenhoek Hospital—Netherlands Cancer Institute (NCI) and the Amsterdam University Medical Centre (AUMC). This study was a secondary analysis of two studies that were approved by the local institutional review boards of the NCI and AUMC (institutional review numbers IRBdm19-348 and 2017.543, respectively). All patients signed informed consent forms when enrolled in the original studies, explicitly allowing secondary analysis of their study data.

### 2.2. Data and Outcome

The preoperative parameters that were collected included age, initial prostate-specific antigen (PSA) level, clinical tumour stage (cT), biopsy ISUP Grade Group (bISUP) and number of (positive) biopsy cores. Also, the prostate imaging and reporting data system (PI-RADS) score and radiological tumour stage (mT) were assessed using multi-parametric magnetic resonance imaging (mpMRI). The intensity of PSMA expression, semi-quantitatively expressed as the SUV_max_ of the dominant intraprostatic lesion, and molecular imaging nodal stage (miN stage) were assessed using preoperative PSMA PET/CT, as well as the type of radiotracer used (i.e., [^68^Ga]Ga-PSMA-11 or [^18^F]DCFPyL). The postoperative variables that were collected included the pathological tumour stage (pT), pISUP, pathological nodal stage (pN stage) and surgical margin status. The outcome was biochemical recurrence, defined as a postoperative PSA level of 0.2 ng/mL or higher in the follow-up.

### 2.3. PSMA PET/CT Imaging Protocol

For AUMC patients, PSMA-PET/CT imaging was performed with [^18^F]DCFPyL, a second-generation fluorinated PSMA radiotracer. The scanner used was a Philips Ingenuity (Philips^®^, Amsterdam, The Netherlands/MA, USA) PET/CT system. [^18^F]DCFPyL was synthesized at the AUMC on-site cyclotron facility according to Good Manufacturing Practices and was supplied to the NCI for imaging purposes. At the NCI, PSMA-PET/CT imaging was performed with both [^68^Ga]Ga-PSMA-11 and [^18^F]DCFPyL tracers using a Philips Gemini TF-II or Vereos Digital PET/CT (Philips^®^, Amsterdam, The Netherlands/MA, USA). [^68^Ga]Ga-PSMA-11 was radiolabelled in-house using a fully automated system (Scintomics GmbH, Gräfelfing, Germany). At the NCI, PSMA PET/CT scans were performed 60 min post-injection (p.i.) of both ^68^Ga-PSMA-11 and [^18^F]DCFPyL PET/CT, while [^18^F]DCFPyL PET/CT scans at the AUMC were performed 120 min p.i. [19]. All PET images were combined with either a low-dose CT (120–140 kV, 40–80 mAs) or a diagnostic CT scan (130 kV, 110 mAs) for anatomical correlation and attenuation correction. All PET images were taken according to the EARL standards and were corrected for scatter, decay and random coincidences [20].

### 2.4. Image Interpretation of PSMA-PET/CT Imaging

At both centres, [^18^F]DCFPyL and [^68^Ga]Ga-PSMA-11 PET/CT scans were assessed by experienced nuclear medicine physicians. The PSMA PET/CT scans were reported in accordance with the E-PSMA guidelines [10]. These reports included the primary intraprostatic lesion, and if present, secondary lesions, miT stage and the presence of lymph node (miN1/M1a stage), bone or visceral metastases (miM1b/c stage) [21]. Imaging results were primarily based on visual and semi-quantitative interpretations according to the E-PSMA guidelines [10]. Positive lesions were reported if the [^68^Ga]Ga-PSMA-11/[^18^F]DCFPyL uptake was higher than prostatic background activity (i.e., scores 4 and 5 based on a 5-point scale according to E-PSMA).

### 2.5. Scan Assessment and SUV_max_ Assessment

For semi-quantitative analysis, the maximum standardised uptake value (SUV_max_) was measured in the dominant intraprostatic lesions. These lesions were correlated with available clinical reports describing the prostatic lesions. SUV_max_ was measured according to the E-PSMA criteria and was compliant with the EARL standards [10,20]. Volumes of interest (VOI) were manually drawn of the size of the prostate area, carefully omitting physiological activity from the bladder or urethra. SUV_max_ was automatically calculated in those VOI using available clinical software: Sectra IDS7 v22.1 (Sectra AB, Linköping, Sweden) and Osirix v12.0 MD (Pixmeo SARL, Bernex, Switzerland). To cross-validate both software programs, identical scans from 10 patients were analysed using both programs, with 100% agreement.

Additionally, the tumour-to-blood ratio (TBR) was calculated in all patients since image-based TBR was found to better characterize the tumour than SUV_max_ can in the quantification of [^18^F]DCFPyL PET/CT scans [22]. The TBR was derived by normalizing the mean tumour uptake (Bq/cc) with the arterial blood activity concentration (Bq/cc) in the ascending aorta.

### 2.6. Statistical Analysis

Continuous variables were assessed for normality using histogram analysis and were summarized with median values and the inter-quartile range (IQR). Categorical variables were summarized with frequencies and proportions. Variables with non-normal deviation were log transformed for further analyses. To compare the medians of the SUV_max_ with preoperative and postoperative variables, the Mann–Whitney–Wilcoxon test was used for two groups, and the Kruskal–Wallis test was used for three groups or more. Correlations between non-parametric continuous variables were assessed using Spearman’s Rho. Cox regression was performed to identify variables that were associated with BCR-free survival. The follow-up time in Cox regression analysis was defined as the duration in days from RARP to the occurrence of BCR, or until the last recorded measurement of PSA level if the patient did not experience BCR. Correlated variables were analysed separately in survival analyses, as multi-collinearity reduces the precision of the estimated coefficients, which weakens the statistical power of the models. Kaplan–Meier plots were drawn to show the relationship between the SUV_max_ and BCR in subgroups. For this purpose, the SUV_max_ values were grouped and plotted as an independent value versus the percentage of patients with BCR in 1 and in 2 years as a dependent value. We visually assessed the risk of BCR per increased value of SUV_max_ to determine low-risk and high-risk groups. The log-rank test was used to evaluate if those subgroups significantly differed from each other.

To analyse the predictive value of SUV_max_ for BCR, proportional hazard models were created and compared. Differences and comparative suitability of models were assessed using both the likelihood ratio test and Akaike information criteria (AIC). Time-dependent receiver operating characteristic (ROC) curves were generated to calculate the time-dependent area under the curve (AUC) for the prediction model of BCR. Harrell’s C concordance indices and R-squared values were calculated for the two prediction models. Statistical significance was set at *p* < 0.05. Statistical analysis was performed using IBM SPSS Statistics for Windows^®^ V27 (Armonk, NY, USA: IBM Corp) and RStudio 4.2.1 for Windows^®^. The figures were generated using GraphPad Prism^®^ software (Version 9.3.1 for Windows, GraphPad Software).

## 3. Results

### 3.1. Baseline Characteristics

A total of 464 patients were included in this study. The baseline characteristics are presented in Table 1. All patients received PSMA PET/CT prior to RARP, of whom 238 (51%) patients were scanned with the tracer [^68^Ga]Ga-PSMA-11, and 226 (49%) patients were scanned with [^18^F]DCFPyL. Of the patients who were scanned with [^18^F]DCFPyL, 106 (47%) patients were scanned 60 min post-radiotracer injection ([^18^F]DCFPyL-60), and 120 (53%) patients were scanned 120 min post-injection ([^18^F]DCFPyL-120). Patients who were scanned with [^68^Ga]Ga-PSMA-11 received a median tracer dose of 99 MBq (IQR 93–105), [^18^F]DCFPyL-60 patients received a median tracer dose of 200 MBq (IQR 192–217) and [^18^F]DCFPyL-120 patients received a median dose of 312 MBq (IQR 301–321). The median age was 68 years (IQR 64–73). The median follow-up time was 24 months (IQR 12–37). BCR was found in 141 patients (30.4%) (57/226 in [^18^F]DCFPyL and 84/238 in [^68^Ga]Ga-PSMA-11 PET/CT scans). A total of 360 patients had ≥1 year PSA follow-up, and 246 patients had ≥2 years follow-up. The one- and two-year BCR-free survival rates were 82.3% and 77.2%, respectively.

### 3.2. Association with Biochemical Recurrence (BCR) of Disease

The preoperative variables that were associated with BCR on univariate Cox regression analysis were: the preoperative Log-2-transformed PSA (Log_2_PSA) (*p* = 0.008), bISUP (*p* < 0.001), mT stage (*p* < 0.001), PI-RADS score (*p* = 0.008), Log_2_SUV_max_ of the entire cohort (*p* < 0.001) as well as the Log_2_SUV_max_ of [^68^Ga]Ga-PSMA-11 (*p* < 0.001) and [^18^F]DCFPyL cohorts (*p* = 0.01), separately (Table 2). The highest hazard ratios (HR) for BCR were seen in patients with mT3b or a PI-RADS score of five (HR 3.83 (95% CI 2.51–5.83), *p* < 0.001 and 2.68 (95% CI 1.09–6.57), *p* < 0.03), respectively).

Multivariate Cox regression analysis for BCR accounting for the preoperative variables (Log_2_PSA, bISUP, mT stage, PI-RADS score and Log_2_SUV_max_) of the entire cohort showed that Log_2_SUV_max_ was not independently associated with BCR (HR 1.20 (95%-CI 0.99–1.45), *p =* 0.06). However, Log_2_PSA, bISUP and mT were found to be independently associated with BCR (*p =* 0.03, *p* < 0.001, *p* < 0.001, respectively) (Table 3).

Multivariate Cox regression analyses accounting for the same co-variates were performed on the various radiotracer groups. Notably, only the [^68^Ga]Ga-PSMA-11 and [^18^F]DCFPyL-120 PET/CT cohorts showed a significant independent association between Log_2_SUV_max_ and BCR (HR 1.32 (95%-CI 1.01–1.72), *p =* 0.04 and HR 1.55 (95%-CI 1.03–2.34), respectively. This association was not found in the [^18^F]DCFPyL-60 patients (HR 0.92 (95%-CI 0.56–1.50), *p* = 0.72. Additional results concerning the subgroups are presented in Table 3.

**Table 3 diagnostics-13-02343-t003:** Multivariate Cox regression analyses for biochemical recurrence after RARP, preoperative variables with SUV_max_.

	All Scans *N =* 464		[68Ga]Ga-PSMA-11 *N =* 238		[^18^F]DCFPyL 120 min *N =* 120		[^18^F]DCFPyL 60 min *N =* 106	
	HR (95% CI)	*p*-Value	HR (95% CI)	*p*-Value	HR (95% CI)	*p*-Value	HR (95% CI)	*p*-Value
Log_2_PSA	1.18 (1.02–1.37)	0.03	1.10 (0.91–1.33)	0.32	1.29 (0.85–1.96)	0.24	1.35 (0.87–2.10)	0.18
Biopsy ISUP Grade Group ^a^		<0.001		<0.001		0.29		0.16
1–2	ref	ref	ref	ref	ref	ref	ref	ref
3	1.24 (0.69–2.24)	0.47	1.77 (0.76–4.12)	0.19	0.61 (0.19–1.94)	0.40	1.48 (0.37–5.99)	0.58
4–5	2.88 (1.82–4.56)	<0.001	4.34 (2.21–8.53)	<0.001	1.37 (0.54–3.45)	0.50	2.81 (0.90–8.80)	0.08
mT		<0.001		<0.001		0.10		0.04
mT0–2	ref	ref	ref	ref	ref	ref	ref	ref
mT3a	1.47 (0.94–2.31)	0.09	1.70 (0.95–3.07)	0.08	2.43 (0.73–8.04)	0.15	1.61 (0.53–4.91)	0.41
mT3b	3.68 (2.30–5.90)	<0.001	3.85 (2.20–6.75)	<0.001	6.55 (1.37–31.24)	0.02	5.00 (1.39–17.97)	0.01
Unknown	2.63 (0.98–7.02)	0.05	*		3.62 (0.89–14.79)	0.07		
PI-RADS-score		0.50		0.40		0.72		0.34
0–3	ref	ref	ref	ref	ref	ref		
4	1.36 (0.51–3.61)	0.54	1.38 (0.38–4.97)	0.62	1.89 (0.33–10.89)	0.48	ref **	ref **
5	1.61 (0.64–4.08)	0.31	1.79 (0.54–5.90)	0.34	1.25 (0.23–6.71)	0.79	0.99 (0.28–3.52)	0.98
Unknown	0.89 (0.26–3.04)	0.85	0.43 (0.04–4.34)	0.47	0.76 (0.14–4.24)	0.75	3.61 (0.56–23.22)	0.18
Log_2_SUV_max_	1.20 (0.99–1.45)	0.06	1.32 (1.01–1.72)	0.04	1.55 (1.03–2.34)	0.04	0.92 (0.56–1.50)	0.72

RARP = robot-assisted radical prostatectomy; SUV_max_ = semi-quantitative measurement of PSMA expression in maximum Standardised Uptake Values; [^18^F]DCFPyL (xx)min = patient scanned after time in minutes after DCFPyL injection; HR= hazard ratio; CI= confidence interval; Log_2_(XX) = log-2 transformed variable; PSA= prostate specific antigen; ISUP = International Society of Urological Pathology Grade Group; mT-stage= radiological tumour stage on MRI; PI-RADS score = prostate imaging reporting and data system; PSMA PET/CT = Prostate Specific Membrane Antigen Positron Emission Tomography/Computerized Tomography. * One patient with missing mT-stage was excluded from the analysis; ** Due to lack of PIRADS 0–3 patients in this group, multivariate analysis of this coefficient was not possible, therefore the new groups where: PIRADS 0–4, PIRADS 5 and unknown. ^a^ ISUP Grade Group = International Society of Urological Pathology grading system [23]: ISUP 1 = Gleason score 3 + 3 = 6; ISUP 2 = Gleason score 3 + 4 = 7; ISUP 3 = Gleason score 4 + 3 = 7; ISUP 4 = Gleason score 4 + 4 = 8/Gleason score 3 + 5 = 8/Gleason score 5 + 3 = 8; ISUP 5 = Gleason score 4 + 5 = 9/Gleason score 5 + 4 = 9/Gleason score 5 + 5 = 10. Using pathological variables (pT, pN, pISUP and surgical margin status) as co-variates in the multivariate Cox regression, Log_2_SUV_max_ was not an independent predictor of BCR (*p* = 0.77). As displayed in Table 4, Log_2_SUV_max_ was not an independent predictor of BCR in the individual radiotracer groups, as well as the [^68^Ga]Ga-PSMA-11 (*p* = 0.14), [^18^F]DCFPyL-120 (*p* = 0.93) and [^18^F]DCFPyL-60 (*p* = 0.10) groups. However, analysing the entire cohort, bISUP (*p* < 0.001), pT (*p* < 0.001) and positive surgical margins (R1) (HR 1.84 (95% CI 1.31–2.67), *p* < 0.001) showed to be significant predictors of BCR. Additional results pertaining to the subgroups can be found in Table 4.

### 3.3. Level of SUV_max_ and Biochemical Recurrence (BCR) of Disease

Analysing the entire cohort, median SUV_max_ was significantly higher in patients who experienced BCR after one and two years of follow-up compared to that of the BCR-free patients (12.02 vs. 7.69, *p* < 0.001 and 11.18 vs. 7.72, *p* = 0.002, respectively). This significant difference was also found via [^68^Ga]Ga-PSMA-11 PET/CT scans (11.99 vs. 7.69, *p* = 0.004 and 11.57 vs. 7.55, *p* = 0.006), as shown in Figure 1. Nevertheless, in the patients scanned with [^18^F]DCFPyL-120, the median SUV_max_ was significantly higher in the BCR patients after one year of follow-up (16.14 vs. 7.69, *p* = 0.007), but not after 2 years of follow-up (14.37 vs. 8.17, *p* = 0.06). In the [^18^F]DCFPyL-60 group, no significant differences in the median SUV_max_ were found (12.32 vs. 7.52, *p* = 0.10 and 6.59 vs. 7.81, *p* = 0.61).

Data presented per radiotracer group: [^18^F]DCFPyL scanned 60 min post-injection ([^18^F]DCFPyL-60), [^18^F]DCFPyL scanned 120 min post-injection ([^18^F]DCFPyL-120) and [^68^Ga]Ga-PSMA-11 after 1 year of follow-up and 2 years of follow-up

The previously mentioned SUV_max_ low- and high- risk groups for BCR were visually assessed, characterising an SUV_max_ ≤ 10 as low risk and an SUV_max_ > 10 as high risk for developing BCR (Appendix A). The analysis of all subgroups in the included cohort demonstrated that SUV_max_ > 10 was a significant predictor for BCR (*p* < 0.001) (Figure 2). SUV_max_ > 10 also showed to be a significant predicting factor for BCR in subgroups with bISUP 3–5, mT3 and an EAU classification high risk for BCR (*p* < 0.001, *p* = 0.002, *p* = 0.01, *p* = 0.004, respectively), but not in patients with mT2 (*p* = 0.07), or patients with EAU intermediate risk for BCR (*p* = 0.06). SUV_max_ > 10 was associated with BCR in univariate cox regression analysis as well (HR 1.87 (95%-CI 1.34–2.60), *p* ≤ 0.001).

Data are presented in graphs comparing the maximum standardised uptake values (SUV_max_) of the dominant intraprostatic lesion from below 10 to a value of 10 or higher in both the whole cohort as well as subgroups of the preoperative variables.

### 3.4. SUV_max_ in a Prediction Model for Biochemical Recurrence (BCR) of Disease

To analyse the added predictive value of the SUV_max_, a baseline predictive model was created, incorporating the preoperative variables, log_2_PSA, bISUP, mT and the PI-RADS score. This model was subsequently compared with a new prediction model, wherein log_2_SUV_max_ was added to the variables of the baseline model. The inclusion of SUV_max_ improved the AIC value of the prediction model (1502 compared to 1498) and was significantly different from the baseline model (*p* = 0.02). After the addition of the SUV_max_ to the baseline model, higher time-dependent AUC values were found at every time point compared to those of the baseline model, as presented in Table 5. Additionally, the Harrel C concordance index was calculated for the baseline prediction model, as well as the new model, yielding nearly similar results of 68.2% and 69.7%, respectively. Similarly, the R-squared values of the two models showed marginal improvements: 11.9% vs. 13.0%.

### 3.5. Association of Tumour-to-Blood Ratio (TBR) with Biochemical Recurrence (BCR) of Disease

Multivariate Cox regression analysis for BCR accounting for preoperative variables (Log_2_PSA, bISUP, mT stage and PI-RADS score) and Log_2_TBR was performed for the entire cohort, as well as the different tracer protocol groups. Log_2_TBR was shown to be an independent predictor for BCR in patients who underwent [^18^F]DCFPyL-120 scanning (HR 5.14 (95%-CI 1.29–20.47), *p* = 0.02), but not in the [^18^F]DCFPyL-60 patients (HR 1.04 (95%-CI 0.33–3.31), *p* = 0.95) (Table 6).

## 4. Discussion

The aim of this study was to investigate whether the PSMA expression of the dominant intraprostatic lesion, defined as SUV_max_ via primary staging PSMA PET/CT, was associated with BCR in patients with primary PCa prior to RARP. Two commonly used radiolabelled PSMA tracers were analysed: [^68^Ga]Ga-PSMA-11 and [^18^F]DCFPyL. We found that SUV_max_ was an independent significant predictor for BCR in patients who undergone [^68^Ga]Ga-PSMA-11 or [^18^F]DCFPyL-120 PET/CT. In addition, SUV_max_ > 10 was shown to be significantly associated with BCR in the entire patient cohort and for individual subgroups: bISUP 3–5, mT3 and patients with the EAU classification of a high risk for BCR.

Due to differences in the scan protocols, and thereby, differences in PSMA tracer uptake, the [^18^F]DCFPyL PET/CT scans were divided into two groups based on the tracer incubation time: 60 min ([^68^Ga]Ga-PSMA-11 and [^18^F]DCFPyL-60) or 120 min ([^18^F]DCFPyL-120) and were evaluated separately. Via univariable analysis, the Log_2_SUV_max_ of the dominant intraprostatic lesion of the entire cohort as well as the Log_2_SUV_max_ of the individual tracer subgroups were shown to be significant predictors for BCR. More importantly, via multivariate analysis, to assess multiple preoperative variables for their ability to predict BCR after surgery, the Log_2_SUV_max_ was shown to be an independent significant predictor for the development of BCR in [^68^Ga]Ga-PSMA-11 and [^18^F]DCFPyL-120 PET/CT scans.

To our knowledge, this study is the first report to describe the association between SUV_max_ and BCR for both [^68^Ga]Ga-PSMA-11 and [^18^F]DCFPyL tracers. Moreover, this is the first study reporting the predictive value of SUV_max_ for BCR in [^18^F]DCFPyL PET/CT scans. Therefore, our findings of SUV_max_ being an independent predictor for BCR in [^18^F]DCFPyL-120 PET/CT scans cannot be compared to other studies. However, previous studies reported on an association between SUV_max_ and BCR in [^68^Ga]Ga-PSMA-11 PET/CT scans [11,14,15]. Roberts et al. published a retrospective analysis of 71 patients who received [^68^Ga]Ga-PSMA-11 PET/CT prior to RARP and showed a 5.5 fold increase in the hazard for BCR in patients with SUV_max_ > 8 compared to SUV_max_ < 8 via multivariate analysis [11].

Furthermore, a retrospective analysis of 186 patients with primary PCa who underwent [^68^Ga]Ga-PSMA-11 PET/CT prior to surgery by Wang et al. [14] showed that patients with a localised disease and a higher SUVmax had a less favourable BCR-free survival rate (*p* = 0.02). In addition, a recent study by Roberts et al. [15] examining a larger cohort of 848 men who underwent [^68^Ga]Ga-PSMA-11 PET/CT scans demonstrated this outcome as well (*p* < 0.001). The SUV_max_ in [^68^Ga]Ga-PSMA-11 PET/CT scans was, therefore, in line with our results, which indicate that it is an independent predictor for BCR. It is important to underline that the independent added value of SUV_max_ for the prediction of BCR is often explained by the strong correlation between the SUV_max_ and tumour grade [11,12,25], resulting in a higher risk for developing BCR.

A possible explanation for the non-significant correlations and associations with BCR in the [^18^F]DCFPyL-60 group might be related to differences in tracer incubation compared to that of the [^18^F]DCFPyL-120 patients [19]. Previous studies demonstrated a significant rise in tumour tracer uptake between 60 and 120 min after the tracer injection, leading to a higher tumour detection rate in [^18^F]DCFPyL-120 patients [19,22]. Therefore, SUV_max_ might not be suitable for universal application across different scan protocols with different tracer uptake intervals, without considering a translation factor and tracer decay.

Another notable difference in the scan protocols was the administered tracer dose in the two [^18^F]DCFPyL scan groups. In the [^18^F]DCFPyL-120 group, a median tracer dose of 312 MBq was injected, compared to 200 MBq in patients who were scanned after 60 min. However, the injected tracer doses are accounted for in the SUV_max_ calculation; therefore, we expect that those differences in tracer dosage have a negligible impact on inter-group SUV_max_ variability. Moreover, different PET/CT scanners were used in this study, with all [^18^F]DCFPyL-120 PET/CT scans performed on the same PET/CT scanner, while [^18^F]DCFPyL-60 and [^68^Ga]Ga-PSMA-11 PET/CT scans were performed interchangeably using two different scanners. The use of different scanners may introduce some variability in the SUV_max_ values; although, this variability is limited for EARL-accredited scanners [26]. Further investigation is required to determine if the SUV_max_ can be directly applied to various scan protocols, PET/CT scanners, or if the implementation of a correction factor is necessary.

Previous research by Jansen et al. suggested that studying the TBR is a superior simplified method for determining the PSMA tracer intensity expression in the SUV_max_ in [^18^F]DCFPyL PET/CT scans [22]. Therefore, we repeated our multivariate analysis, including the TBR instead of the SUV_max_. This yielded an even more significant association with BCR for [^18^F]DCFPyL-120 and a slightly negative, non-significant association for [^18^F]DCFPyL-60. These outcomes align with previous pharmacokinetic results, showing that the TBR is highly influenced by the interval of tracer uptake. We therefore underline the importance of scanning patients 120 min after an injection when using the [^18^F]DCFPyL tracer.

Another possible explanation for the differences between the scan groups might be the smaller sample size of the two individual [^18^F]DCFPyL scan groups compared to that of the [^68^Ga]Ga-PSMA-11 group, as no major differences in surgical demographics between the groups were present.

In this study, we observed that patients with high SUV_max_ values (SUV_max_ > 10) and less-favourable preoperative variables had significantly higher BCR rates compared to those of the patients with low SUV_max_ values (SUV_max_ ≤ 10) in the same subgroups. The identification of factors that could aid in predicting survival outcomes is crucial, since postoperative disease recurrence is challenging to predict [5]. Therefore, we believe that incorporating the SUV_max_/TBR of primary PET/CT scans into preoperative prediction models could be of added value.

Furthermore, we found that the prediction model incorporating both preoperative variables and SUV_max_ is superior to the baseline prediction model that only included preoperative variables. As a result, slightly higher values for the area under the curve (AUC) were observed at various time points. Also, slightly higher R-squared values and C-indexes were found. Notably, a significant improvement of the AIC was found when SUV_max_ was added to the baseline prediction model. These findings are in line with literature, as PSMA PET/CT is increasingly being incorporated into preoperative risk stratification models. Qui et al. [27] retrospectively investigated the prognostic role of preoperative [^68^Ga]Ga-PSMA-11 PET/CT scans in predicting BCR after RARP. They evaluated 77 patients after 2 years of follow-up and demonstrated the superior discriminative ability of a risk model incorporating SUV_max_ compared to that of the CAPRA and D’Amico models. The exact additional value of SUV_max_ in these models needs to be explored further in future, larger cohort studies.

A strength of our study was that VOI’s delineation of the dominant intraprostatic lesions was performed by one expert. Therefore, no inter-observer variability occurred. Moreover, delineating VOI, and thus, calculating the SUV_max_ is a simple action and can, therefore, easily be implemented in current clinical practice. A disadvantage of this method is that manual VOI alteration is necessary when the suspected dominant intraprostatic lesion is close to the bladder wall to rule out bladder activity interference. To solve this issue, Artificial intelligence tools might play an important role in the future.

This study has several limitations that need to be acknowledged. Firstly, it was conducted in a retrospective setting, which introduces the possibility of selection bias. Secondly, we did not incorporate the total tumour volume (TTV) as a variable in this study. Considering that the SUV_max_ is also influenced by tumour volume, the inclusion of the TTV would have provided more accurate SUV_max_ values [22], since the TTV also has prognostic abilities [24]. Instead, we demonstrated that utilizing the PI-RADS score as a surrogate for tumour volume, wherein a tumour volume of ≥15mm via MRI differentiates between a PI-RADS 4 or 5 lesion [28], yielded satisfactory results. Thirdly, the bi-centric setup resulted in different scan protocols, thereby leading to variations in the obtained results, as previously stated.

Based on the present findings, SUV_max_ could contribute to the prediction of BCR in patients with primary PCa before undergoing RARP. Future, prospective studies are required to re-evaluate whether the SUV_max_ of the dominant intraprostatic lesion can be considered as an addition to prediction models. Moreover, larger studies are needed to evaluate the translatability of SUV_max_ to different scan protocols.

## 5. Conclusions

This study evaluated the prognostic value of SUV_max_ for the biochemical recurrence of disease after RARP in patients with PCa undergoing [^68^Ga]Ga-PSMA-11 and [^18^F]DCFPyL PET/CT. SUV_max_ proved to be an independent predictive factor for BCR in patients with primary PCa and in patients who underwent [^68^Ga]Ga-PSMA-11 PET/CT or [^18^F]DCFPyL PET/CT and were scanned 120 min after tracer injection. In these tracer groups, the median SUV_max_ was significantly higher in patients with BCR after 1 year of follow-up. Moreover, SUV_max_ > 10 was a significant predictor for BCR. Therefore, the SUV_max_ in PSMA PET/CT may be of added value to risk stratification models for BCR in patients with primary PCa.

## Figures and Tables

**Figure 1 diagnostics-13-02343-f001:**
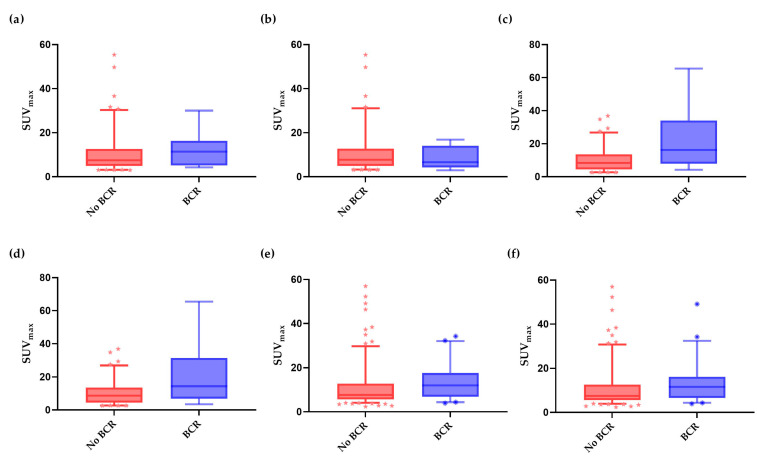
Boxplots displaying differences in maximum standardised uptake values (SUV_max_) of patients who experienced biochemical recurrence (BCR) and patients who had no BCR. (**a**) [^18^F]DCFPyL-60 SUV_max_, 1 year follow-up (median SUV_max_ 7.52 vs. 12.32, *p* = 0.10), *n* = 106; (**b**) [^18^F]DCFPyL-60 SUV_max_, 2 years follow-up (median SUV_max_ 7.81 vs. 6.59, *p* = 0.61), *n* = 101; (**c**) [^18^F]DCFPyL-120 SUVmax, 1 year follow-up (median SUV_max_ 7.96 vs. 16.14, *p* = 0.007), *n* = 110; (**d**) [^18^F]DCFPyL-120 SUVmax, 2 years follow-up (median SUV_max_ 8.17 vs. 14.37, *p* = 0.06), *n* = 107; (**e**) [^68^Ga]Ga-PSMA SUV_max_,1 year follow-up (median SUV_max_ 7.69 vs. 11.99, *p* = 0.004), *n* = 236; (**f**) [^68^Ga]Ga-PSMA SUV_max_, 2 years follow-up (median SUV_max_ 7.55 vs. 11.57, *p* = 0.006), *n* = 230.

**Figure 2 diagnostics-13-02343-f002:**
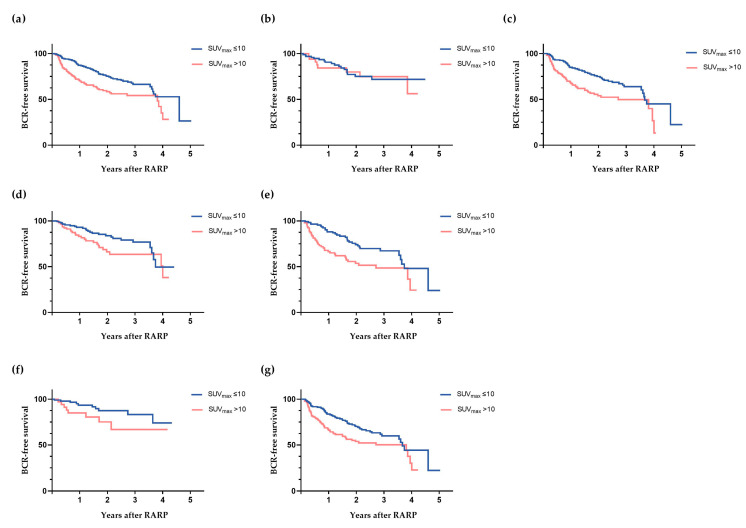
Kaplan–Meier curves showing the probability of biochemical-recurrence-free survival (BCR-free survival) in patients years after robot-assisted radical prostatectomy (RARP). (**a**) All patients (Log-Rank *p* < 0.001), *n* = 464; (**b**) Biopsy ISUP-grade (bISUP) 1–2 (Log-Rank *p* = 0.76), *n* = 129; (**c**) bISUP 3–5 (Log-Rank *p* < 0.001), *n* = 335; (**d**) Magnetic resonance imaging tumour stage 2 (mT2) (Log-Rank *p* = 0.07), *n* = 237; (**e**) mT3 (Log-Rank *p* = 0.002), *n* = 209; (**f**) European Association of Urology (EAU) intermediate risk for BCR (Log-Rank *p* = 0.06), *n* = 128; (**g**) EAU high risk for BCR (local + locally advanced) (Log-Rank *p* = 0.004), *n* = 317.

**Table 1 diagnostics-13-02343-t001:** Baseline characteristics of patients undergoing PSMA PET/CT for initial staging.

Median (IQR)		[^18^F]DCFPyL (*N* = 226)	[^68^Ga]Ga-PSMA-11 (*N* = 238)
Age (years)	Median (IQR)	68 (63–73)	68 (64–73)
Initial PSA level (ng/mL)	Median (IQR)	9.7 (6.8–17)	9.6 (6.7–18)
Positive biopsy cores (% of total cores)	Median (IQR)	42 (29–63)	50 (25–67)
Intraprostatic SUV_max_	Median (IQR)	8.2 (4.9–13.7)	8.0 (5.8–14.4)
Follow-up time (months)	Median (IQR)	19 (9–31)	32 (19–41.3)
***N* (%)**			
BCR	*N* (%)		
NO BCR		169 (75%)	154 (65%)
BCR		57 (25%)	84 (35%)
mT	*N* (%)		
mT0-2		116 (51)	139 (58)
mT3a		55 (24)	56 (24)
mT3b		18 (8.0)	42 (18)
unknown		37 (16)	1 (0.4)
Biopsy ISUP Grade Group ^a^	*N* (%)		
1		10 (4.4)	18 (7.6)
2		62 (27)	39 (16)
3		65 (29)	61 (26)
4		56 (25)	75 (32)
5		33 (15)	45 (19)
EAU-risk classification	*N* (%)		
low		1 (0.4)	2 (0.8)
intermediate		77 (34)	51 (21)
high		110 (49)	134 (56)
locally advanced		24 (11)	49 (21)
unknown		14 (6.2)	2 (0.8)
miN	*N* (%)		
miN0		199 (88)	214 (90)
miN1		27 (12)	24 (10)

PSMA PET/CT = Prostate-Specific Membrane Antigen Positron Emission Tomography/Computerized Tomography; *N* = number of patients; IQR = Interquartile range; PSA = prostate-specific antigen; BCR = biochemical recurrence; mT stage = radiological tumour stage on MRI; ISUP = International Society of Urological Pathology Grade Group; EAU = European Association of Urology risk classification [3]. cT = clinical tumour stage; miN = molecular imaging lymph node stage. ^a^ ISUP Grade Group = International Society of Urological Pathology grading system [23]: ISUP 1 = Gleason score 3 + 3 = 6; ISUP 2 = Gleason score 3 + 4 = 7; ISUP 3 = Gleason score 4 + 3 = 7; ISUP 4 = Gleason score 4 + 4 = 8/Gleason score 3 + 5 = 8/Gleason score 5 + 3 = 8; ISUP 5 = Gleason score 4 + 5 = 9/Gleason score 5 + 4 = 9/Gleason score 5 + 5 = 10.

**Table 2 diagnostics-13-02343-t002:** Univariable Cox regression analyses for biochemical recurrence after RARP.

	HR (95% CI)	*p*-Value
Log_2_PSA (ng/mL)	1.22 (1.05–1.41)	0.01
Log_2_(positive biopsy cores) (%)	1.85 (0.96–3.59)	0.07
Biopsy ISUP Grade Group ^a^		<0.001
1–2	ref	ref
3	0.95 (0.54–1.68)	0.86
4–5	2.40 (1.56–3.72)	<0.001
mT		<0.001
mT0–2	ref	ref
mT3a	1.57 (1.03–2.39)	0.04
mT3b	3.83 (2.51–5.83)	<0.001
Unknown	1.63 (0.89–2.98)	0.11
PI-RADS-score		0.01
0–3	ref	ref
4	1.43 (0.54–3.76)	0.48
5	2.68 (1.09–6.57)	0.03
Unknown	1.73 (0.63–4.78)	0.29
Log2SUV_max_ all	1.42 (1.21–1.68)	<0.001
Log2SUV_max_ [^68^Ga]Ga-PSMA-11	1.48 (1.18–1.85)	<0.001
Log2SUV_max_ [^18^F]DCFPyL	1.37 (1.07–1.75)	0.01
SUV_max_ > 10	1.87 (1.34–2.60)	<0.001

RARP = Robot-Assisted Radical Prostatectomy; HR = hazard ratio; Log2(XX) = log-2-transformed variable; iPSA = initial prostate-specific antigen; ISUP = International Society of Urological Pathology Grade Group; mT stage = radiological tumour stage on MRI; PI-RADS score = prostate imaging reporting and data system; PSMA PET/CT = Prostate-Specific Membrane Antigen Positron Emission Tomography/Computerized Tomography; SUV_max_ = semi-quantitative measurement of PSMA expression defined as maximum Standardised Uptake Values. ^a^ ISUP Grade Group = International Society of Urological Pathology grading system [23]: ISUP 1 = Gleason score 3 + 3 = 6; ISUP 2 = Gleason score 3 + 4 = 7; ISUP 3 = Gleason score 4 + 3 = 7; ISUP 4 = Gleason score 4 + 4 = 8/Gleason score 3 + 5 = 8/Gleason score 5 + 3 = 8; ISUP 5 = Gleason score 4 + 5 = 9/Gleason score 5 + 4 = 9/Gleason score 5 + 5 = 10.

**Table 4 diagnostics-13-02343-t004:** Multivariate Cox regression analyses for biochemical recurrence after RARP, postoperative variables with SUV_max_.

	All Scans		[^68^Ga]Ga-PSMA-11		^18^F-DCFPyL 120 min		^18^F-DCFPyL 60 min	
	HR (95% CI)	*p*-Value	HR (95% CI)	*p*-Value	HR (95% CI)	*p*-Value	HR (95% CI)	*p*-Value
Pathology T-stage		<0.001		<0.001		<0.001		0.11
2	ref	ref	ref	ref	ref	ref	ref	ref
3a	1.88 (1.18–3.00)	0.008	1.84 (0.99–3.42)	0.05	1.79 (0.63–5.11)	0.27	1.13 (0.37–3.49)	0.83
3b	3.86 (2.36–6.31)	<0.001	3.58 (1.93–6.63)	<0.001	9.64 (2.83–32.79)	<0.001	3.52 (0.96–12.92)	0.058
Pathology ISUP-grade ^a^		<0.001		0.006		0.02		0.04
2	ref	ref	ref	ref	ref	ref	ref	ref
3	2.97 (1.73–5.10)	<0.001	2.36 (1.23–4.55)	0.01	3.43 (0.70–16.70)	0.13	5.45 (1.35–22.02)	0.02
4	3.41 (1.72–6.75)	<0.001	2.92 (1.32–6.45)	0.008	10.25 (1.47–71.60)	0.02	1.87 (0.18–19.40)	0.60
5	4.79 (2.63–8.70)	<0.001	3.69 (1.75–7.78)	<0.001	8.93 (1.71–46.54)	0.009	9.31 (1.87–46.27)	0.006
Positive surgical margins	1.87 (1.31–2.67)	<0.001	2.14 (1.34–3.41)	0.001	2.27 (1.08–4.75)	0.03	1.56 (0.57–4.28)	0.39
Log_2_SUV_max_	1.03 (0.85–1.24)	0.77	1.21 (0.94–1.55)	0.14	1.02 (0.69–1.50)	0.93	0.59 (0.32–1.11)	0.10

RARP= robot-assisted radical prostatectomy; SUVmax = semi-quantitative measurement of PSMA expression in maximum Standardised Uptake Values; [18F]DCFPyL (xx)min = patient scanned after time in minutes after DCFPyL injection; HR= hazard ratio; CI= confidence interval; T-stage= tumour stage; ISUP = International Society of Urological Pathology Grade Group; Log2(XX) = log-2 transformed variable. ^a^ ISUP = International Society of Urological Pathology grading system [23]: ISUP 1 = Gleason score 3 + 3 = 6; ISUP 2 = Gleason score 3 + 4 = 7; ISUP 3 = Gleason score 4 + 3 = 7; ISUP 4 = Gleason score 4 + 4 = 8/Gleason score 3 + 5 = 8/Gleason score 5 + 3 = 8; ISUP 5 = Gleason score 4 + 5 = 9/Gleason score 5 + 4 = 9/Gleason score 5 + 5 = 10.

**Table 5 diagnostics-13-02343-t005:** AUC analyses of time-dependent ROC curves for prediction models for biochemical recurrence after RARP.

	0.5 Years	1 Year	1.5 Years	2 Years
**Baseline model** Consisting of preoperative variables: log_2_PSA, biopsy ISUP, mT, PI-RADS score	AUC 65.1%	AUC 64.1%	AUC 64.0%	AUC 63.3%
**New model** Consisting of preoperative variables + log_2_SUV_max_	AUC 66.7%	AUC 65.8%	AUC 65.5%	AUC 64.2%

RARP = robot-assisted radical prostatectomy; AUC = area under the curve; Log_2_(XX) = log-2-transformed variable; PSA = prostate-specific antigen; ISUP = International Society of Urological Pathology Grade Group; mT stage = radiological tumour stage on MRI; PI-RADS score = prostate imaging reporting and data system; PSMA PET/CT = Prostate-Specific Membrane Antigen Positron Emission Tomography/Computerized Tomography; SUV_max_ = semi-quantitative measurement of PSMA expression in maximum Standardised Uptake Values.

**Table 6 diagnostics-13-02343-t006:** Multivariate Cox regression analyses for biochemical recurrence after RARP, preoperative variables with TBR.

	All Scans *N =* 464		[68Ga]Ga-PSMA-11 *N =* 238		[^18^F]DCFPyL 120 min *N =* 120		[^18^F]DCFPyL 60min *N =* 106	
	HR (95% CI)	*p*-Value	HR (95% CI)	*p*-Value	HR (95% CI)	*p*-Value	HR (95% CI)	*p*-Value
Log_2_PSA	1.20 (1.03–1.38)	0.02	1.14 (0.96–1.37)	0.15	1.27 (0.83–1.92)	0.27	1.33 (0.86–2.07)	0.20
Biopsy ISUP Grade Group ^a^		<0.001		<0.001		0.36		0.22
1–2	ref	ref	ref	ref	ref	ref	ref	ref
3	1.26 (0.70–2.27)	0.44	1.77 (0.76–4.12)	0.19	0.63 (0.20–1.99)	0.43	1.42 (0.36–5.68)	0.62
4–5	2.92 (1.85–4.61)	<0.001	4.34 (2.21–8.52)	<0.001	1.33 (0.54–3.30)	0.54	2.58 (0.85–7.84)	0.09
mT		<0.001		<0.001		0.13		0.04
mT0–2	ref	ref	ref	ref	ref	ref	ref	ref
mT3a	1.48 (0.94–2.32)	0.09	1.60 (0.89–2.86)	0.11	2.15 (0.66–6.98)	0.20	1.57 (0.51–4.87)	0.43
mT3b	3.75 (2.34–6.01)	<0.001	3.84 (2.19–6.73)	<0.001	5.54 (1.19–25.77)	0.03	4.86 (1.36–17.34)	0.01
Unknown	2.46 (0.93–6.51)	0.07	*		3.33 (0.85–12.96)	0.08		
PI-RADS-score		0.52		0.40		0.69		0.31
0–3	ref	ref	ref	ref	ref	ref		
4	1.30 (0.49–3.47)	0.60	1.19 (0.33–4.25)	0.79	2.03 (0.36–11.52)	0.42	ref **	ref **
5	1.53 (0.60–3.89)	0.37	1.60 (0.48–5.29)	0.44	1.39 (0.26–7.47)	0.70	1.00 (0.28–3.56)	1.00
Unknown	0.83 (0.25–2.83)	0.77	0.35 (0.04–3.47)	0.37	0.80 (0.14–4.48)	0.80	3.82 (0.60–24.21)	0.16
Log_2_TBR	1.69 (0.95–2.89)	0.07	1.62 (0.74–3.53)	0.23	5.14 (1.29–20.47)	0.02	1.04 (0.33–3.31)	0.95

RARP = robot-assisted radical prostatectomy; TBR = tumour to blood ratio; [18F]DCFPyL (xx)min = patient scanned after time in minutes after DCFPyL injection; HR= hazard ratio; CI= confidence interval; Log2(XX) = log-2 transformed variable; PSA= prostate specific antigen; ISUP = International Society of Urological Pathology Grade Group; mT-stage= radiological tumour stage on MRI; PI-RADS score = prostate imaging reporting and data system; PSMA PET/CT = Prostate Specific Membrane Antigen Positron Emission Tomography/Computerized Tomography; SUVmax = semi-quantitative measurement of PSMA expression in maximum Standardised Uptake Values. * One patient with missing mT-stage was excluded from the analysis; ** Due to lack of PIRADS 0–3 patients in this group, multivariate analysis of this coefficient was not possible, therefore the new groups where: PIRADS 0–4, PIRADS 5 and unknown. ^a^ ISUP Grade Group = International Society of Urological Pathology grading system [24]: ISUP 1 = Gleason score 3 + 3 = 6; ISUP 2 = Gleason score 3 + 4 = 7; ISUP 3 = Gleason score 4 + 3 = 7; ISUP 4 = Gleason score 4 + 4 = 8/Gleason score 3 + 5 = 8/Gleason score 5 + 3 = 8; ISUP 5 = Gleason score 4 + 5 = 9/Gleason score 5 + 4 = 9/Gleason score 5 + 5 = 10.

## Data Availability

The data presented in this study are available on request from the corresponding author. The data are not publicly available due to privacy.

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
