# Peer review of "Higher Preoperative Maximum Standardised Uptake Values (SUVmax) Are Associated with Higher Biochemical Recurrence Rates after Robot-Assisted Radical Prostatectomy for [68Ga]Ga-PSMA-11 and [18F]DCFPyL Positron Emission Tomography/Computed Tomography"

_diagnostics, 2023, doi:10.3390/diagnostics13142343_

Round 1

Reviewer 1 Report

Dear Authors,

your manuscript is well-written and the topic is interesting, with nice results. The methods are detailed and precise.  The discussion can be implemented, and I ask you to better discuss the intrinsic limit and value of SUV measures (different scanners, different tracers, doses...) in a study like yours. 

A minor remark: In Figure 2 the Kaplan-Meier curves are a bit blurry and difficult to read. 

Author Response

Dear reviewer, 

We would like to thank you for the kind words and the time invested to review our manuscript. We have addressed all useful comments in the sections below and hope these adjustments have improved the quality of our manuscript. The response to your comments can be found in the PDF file attached.

Reviewer 2 Report

The study of de Bie et al. aims to explore the role of SUVmax values calculated on PSMA PET/CT for prostate cancer staging correlate to the preoperative and postoperative prediction of biochemical recurrence (BCR) after curative treatment. The topic showcases a new input in the field, detailing a still underresearched subject, which could potentially improve the risk stratification methods for BCR in patients diagnosed with prostate cancer. The methods and the results are clearly presented. The discussions are well detailed and sustain the obtained results, and the conclusions are consistent with the evidence and arguments presented. The figures included in the manuscript are well done and accurately illustrate the results. My only suggestion would be improving the Introduction section, by including more details on the two used radiotracer’s (if [18F]DCFPyL has a better affinity for PCa cells than [68Ga]Ga-PSMA-11, the difference between the two etc.) I also suggest using fluorine instead of fluor to describe F18.

Other than that, the manuscript is well written, there are no detected issues on English language, and the cited references are appropriate. I believe that the manuscript can be published after these minor changes.

Author Response

(The authors gave the same response as above.)

Reviewer 3 Report

Prostate-specific membrane antigen (PSMA) is over expressed predominantly (greater than 90%) in prostate cancers (PCa), second leading cause of cancer death in men. Theranostic PSMA radiotracers have been developed for PCa imaging, diagnosis, staging, stratification and therapy. Among which, 68Ga-PSMA-11 and 18F-DCFPyL (now known as flotufolastat fluorine-18) are tandemly approved by FDA as PSMA imaging PET tracers and have been intensively evaluated in clinics. Herein, authors investigated the association between maximum standardized uptake value (SUVmax) of 68Ga-PSMA-11/18F-DCFPyL and biochemical recurrence (BCR) in a multi-center cohort study of 446 patients. Despite of several limitations described in the manuscript, the study concluded that SUVmax of 68Ga-PSMA-11 and 18F-DCFPyL (120 min p.i.) in PCa patients are independent predictive factor for BCR while SUVmax > 10 was a significant predictor for BCR. This provides a useful tool for BCR rates prediction in PCa patients after Robot-3 assisted Radical Prostatectomy and would draw significant interests in PCa diagnosis and therapy. Therefore, I recommend this manuscript to be published in Diagnostics with minor revision (see attached PDF file).

Author Response

(The authors gave the same response as above.)
